# Position: The Artificial Intelligence and Machine Learning Community Should Adopt a More Transparent and Regulated Peer Review Process

**Jing Yang** [* 1 2]

## Abstract

The rapid growth of submissions to top-tier Artificial Intelligence (AI) and Machine Learning (ML) conferences has prompted many venues to transition from closed to open review platforms. Some have fully embraced open peer reviews, allowing public visibility throughout the process, while others adopt hybrid approaches, such as releasing reviews only after final decisions or keeping reviews private despite using open peer review systems. In this work, we analyze the strengths and limitations of these models, highlighting the growing community interest in transparent peer review. To support this discussion, we examine insights from **Paper Copilot** (https://papercopilot.com/), a website launched two years ago to aggregate and analyze AI / ML conference data while engaging a global audience. The site has attracted over 200,000 early-career researchers, particularly those aged 18–34 from 177 countries, many of whom are actively engaged in the peer review process. *Drawing on our findings, this position paper advocates for a more transparent, open, and well-regulated peer review aiming to foster greater community involvement and propel advancements in the field.*

## 1. Introduction

The exponential growth in submissions to top-tier Artificial Intelligence and Machine Learning (AI / ML) conferences has created unprecedented challenges for the academic review process. With submission numbers exceeding 10,000 for AI / ML venues (Weissburg et al., 2024), traditional review practices are under immense pressure to maintain fairness, efficiency, and quality. In response, many conferences have adopted open review platforms, as illustrated in

---

[1]University of Southern California [2]papercopilot.com. Correspondence to: Jing Yang <jingyang.carl.work@gmail.com>.

*Proceedings of the 42nd International Conference on Machine Learning*, Vancouver, Canada. PMLR 267, 2025. Copyright 2025 by the author(s).

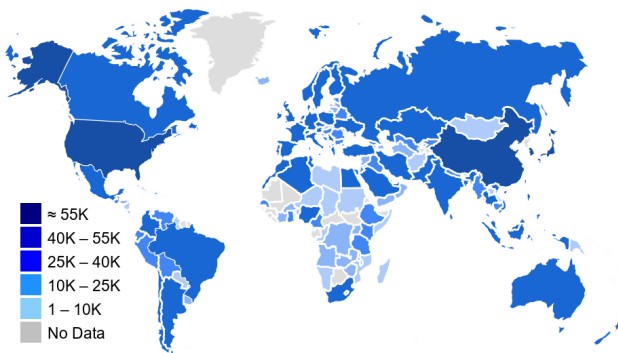

Figure 1: Global 200K+ active users (K = thousands) of Paper Copilot engaging in the usage of open statistics of top AI / ML venues, distributed across 177 countries. The color scale indicates the volume of active users per country, as tracked by Google Analytics (2025) over the past two years. This distribution highlights the community's strong and widespread interest in transparency.

Figure 2. However, the implementation of open peer reviews varies significantly, reflecting diverse decisions by conference organizers. These models—**fully open**, **partially open**, and **closed**—share a common double-blind review framework, where neither authors nor reviewers know each other's identity during the review phase. The key differences lie in the timing and extent of public disclosure of reviews and discussions. Fully open reviews (Ross-Hellauer, 2017) make all content public from the start, partially open reviews disclose reviews after final decisions, and closed reviews do not disclose reviews at all. These differing approaches have sparked debates about their implications for transparency, accountability, and community engagement.

Fully open reviews promote transparency by making review content and discussions accessible to the broader community (Tran et al., 2021; Cortes & Lawrence, 2021; Lawrence, 2022; Beygelzimer et al., 2023; Wang et al., 2023), fostering collaboration and accountability. However, even with double-blind protocols in place, the public nature of fully open reviews can introduce subtle biases or discourage candid feedback from reviewers wary of visibility or potential backlash. In contrast, partially open and closed reviews provide a more private environment, encouraging frank critique

but limiting transparency and broader engagement. These trade-offs raise critical questions about the best practices for academic review processes in rapidly evolving fields like AI and ML, where robust systems are vital to fostering innovation and collaboration.

To explore these dynamics, we publicly launched **Paper Copilot** two years ago—a platform designed to aggregate and analyze data from AI / ML conferences. By sourcing information from official conference websites, review platforms, and community inputs, Paper Copilot tracks engagement throughout the review and decision-making process. Figure 1 presents a global user distribution map derived from Google Analytics (2025), showcasing the diverse geographic reach of Paper Copilot users. This global participation underscores the community's interest in transparency and collaboration within the review process. Through its data aggregation and analysis capabilities, we highlight trends and patterns in review practices, providing valuable insights into how transparency impacts engagement in AI / ML reviewing process.

In this work, we contribute to the ongoing discussion on review transparency in the AI / ML community by:

- Providing open statistics via Paper Copilot, including visualizations of review score distributions, review timelines, and author/affiliation analyses across conferences over the past 3–5 years.

- Presenting quantitative evidence of the community's increasing interest in review transparency, based on two years of engagement data.

- Critically examining the strengths and weaknesses of various review models while advocating for the adoption of a more transparent, open, and regulated peer review process.

**Based on our findings, this position paper advocates for a more transparent, open, and regulated peer review process to enhance community involvement, foster collaboration, and drive progress in the field.**

## 2. Related Works

### 2.1. Open Peer Review

Open peer review (OPR) enhances transparency by publishing reviews, revealing reviewer identities, or enabling public discussions (Ross-Hellauer, 2017; Henriquez, 2023; Wolfram et al., 2020). In AI and ML, OpenReview (OpenReview, 2025) has facilitated OPR, with ICLR pioneering public discourse alongside formal reviews (Wang et al., 2023). Proponents argue that open reviews improve feedback quality, help reviewers refine their assessments (Church

et al., 2024), and enable confidence estimation from review text (Bharti et al., 2022). However, experiments at NeurIPS reveal inconsistencies in peer review (Cortes & Lawrence, 2021; Lawrence, 2022; Beygelzimer et al., 2023), raising concerns about subjective scoring (Xie et al., 2024) and the impact of increasing submissions (Tran et al., 2021). Some studies suggest interventions to reduce uncertainty in reviewer judgments (Chen & Zhang, 2023) or explore author self-assessments as a complement to peer review (Su et al., 2024).

Despite its benefits, OPR within double-blind settings poses challenges. Publishing reviews, even anonymously, may reveal sensitive details or invite targeted criticism (Tran et al., 2021). Computational studies highlight fairness disparities in peer review (Zhang et al., 2022), and alternatives like managing research evaluation on GitHub have been proposed (Takagi, 2022). Broader concerns persist, including whether reviewing efforts align with academic impact (Church et al., 2024) and how best to address systemic biases (Shah, 2022). As NeurIPS discussions occur mid-year and ICLR discussions happen later, the timing of transparency measures may also shape reviewer behavior and decision-making.

### 2.2. Regulations

As OPR evolves, regulatory guidelines ensure integrity, fairness, and privacy (Ross-Hellauer & Görögh, 2019). Some researchers caution that excessive transparency may undermine review quality (Bianchi & Squazzoni, 2022), while others highlight the challenge of balancing confidentiality with open science (Baez, 2002; Dennis et al., 2019).

AI/ML conferences face additional regulatory challenges. Public review platforms can expose researchers to scrutiny or harassment, raising ethical concerns (Wang et al., 2023). AI-powered peer review introduces risks that require human oversight (Seghier, 2024), while plagiarism in review reports and the rise of review mills threaten review integrity (Piniewski et al., 2024; Oviedo-García, 2024; Ezhumalai et al., 2024). To address these risks, researchers advocate for clearer policies on reviewer disclosures, public critique, and misconduct prevention, ensuring transparency strengthens rather than undermines the review process (Kaltenbrunner et al., 2022; Kuznetsov et al., 2024).

## 3. Open Statistics: Paper Copilot

Moving toward a more transparent AI / ML community has become a prominent topic at various venues and within the broader research ecosystem. However, the push for more openness and regulation must be guided by concrete evidence of community needs and interests. Despite growing discussions, there is a lack of quantitative evidence reflect-

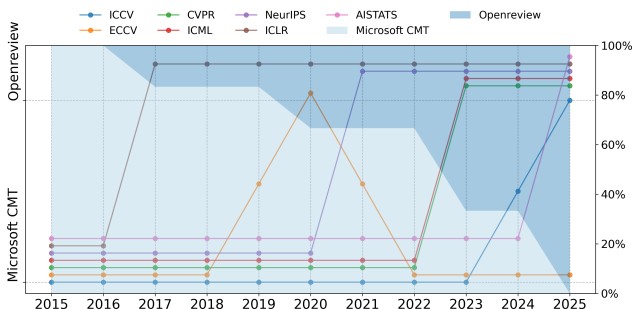

Figure 2: Adoption of Review Platforms among Top-Tier AI / ML Conferences (2015–2025). Data sourced from 'Submission Instructions' or 'Call for Papers' sections on the respective venues' websites.

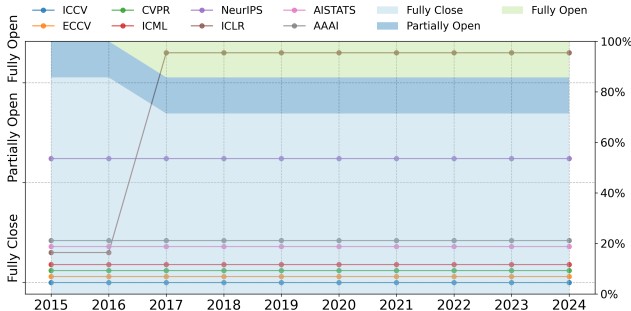

Figure 3: Review Disclosure Preferences among Top-Tier AI / ML Conferences (2015–2024). Definitions of closure categories are detailed in Section 4. 2025 is excluded due to some venues not having announced their preferences yet.

ing the community's true interests and practices around open reviewing. To address this gap, we created *Paper Copilot*—a website designed to deliver research-related services and insights for the AI / ML community.

In this section, we explain how Paper Copilot collects, analyzes, and presents open statistics on review processes. We also discuss our preliminary observations regarding web traffic and user demographics via Google Analytics, setting the stage for the deeper analyses in Section 4, where we reinforce our position that standardized, open, and regulated review process are essential to meet the evolving demands of AI / ML researchers.

### 3.1. Data Collection Methodology

Paper Copilot provides research-related services by gathering and visualizing key metrics from AI / ML conferences. These venues vary in their reviewing models—ranging from choices that expose all review discussions publicly to those that remain fully private. To accommodate these variations, we employ two main strategies for obtaining data:

1. **Automated Retrieval via Public APIs and Site Bots:** When review data are publicly available (e.g., via the OpenReview (2025) API for ICLR), our custom bots retrieve key metrics such as ratings, confidence levels, and reviewer comments. These bots run on a daily schedule, creating a temporal profile that documents how scores and discussions evolve throughout the review cycle.

   Additionally, we enhance our data collection by deploying bots on the official websites of the respective venues. This approach allows us to include descriptive details such as author identities and affiliations while also enabling us to identify and address inconsistencies across data sources.

2. **Community Submissions via Google Forms:** For

partially open or closed-review venues where data are not shared publicly during the review process, we invite authors to voluntarily submit anonymized review information via Google Forms embedded on the Paper Copilot website. This community-driven approach underscores researchers' appetite for transparency even when official policies restrict open peer review data.

Table 1 summarizes the applicability of each review collection method to conferences based on their review disclosure preferences. In total, we processed 10 years of available data from 24 venues across 9 subfields in the field of AI / ML. Over the past yea, we gathered 3,876 valid responses through Community Submissions.

| Venues \ Methods | API | Site Bots | Google Form |
|---|---|---|---|
| Fully Open | ✓ | ✓ | |
| Partially Open | ✓ | ✓ | ✓ |
| Fully Close | | ✓ | ✓ |

Table 1: Review Collection Methods

The collected data from multiple sources is processed using a standardized pipeline to clean, merge, and store it systematically. The resulting datasets are made open-source and are visualized through an interactive frontend. This interface provides insights into review distributions, temporal trends in scores, and basic analytics on authors and affiliations.

### 3.2. Traffic and Engagement Overview

We use Google Analytics (2025) to track page views, session durations, referral sources, and basic demographic details (e.g., user location, device type) for Paper Copilot. Also, the collected data is validated via Matomo (2025). **No personally identifying information is collected, ensuring user privacy.**

Table 2 illustrates that the majority of users arrive via or-

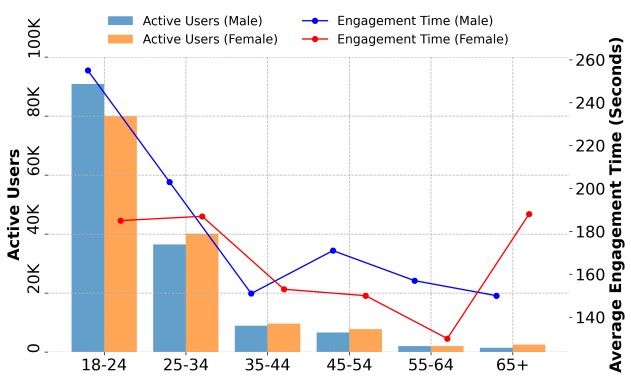

(a) Active Users and Engagement Time by Age and Gender

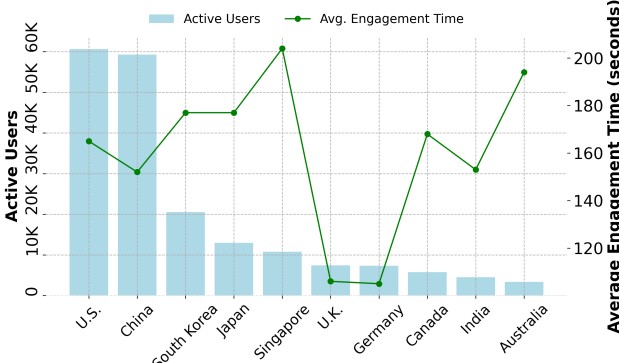

(b) Active Users and Engagement Time by Country

Figure 4: Community Engagement on Paper Copilot: Active Users (K = thousands) visiting the Open statistics and their average engagement time (seconds). (a) Metrics by age and gender. (b) Metrics by country for the top 10 sources of traffic.

| Channel | Percentage | Engaged Users / Events |
|---|---|---|
| Organic Search | 59.9% | 110 K / 1.96 M |
| Direct | 23.9% | 62 K / 0.89 M |
| Referral | 9.4% | 19 K / 0.29 M |
| Organic Social | 7.6% | 14 K / 0.22 M |

Table 2: Traffic distribution across channels, with user engagement (K = thousands) and total events (M = millions) for each source. An event represents any action triggered by a user, such as a click, page view, or scroll, measured via Google Analytics

ganic search (e.g., Google, Bing, Baidu, Yahoo Search), suggesting that researchers actively seek information on review processes and publication statistics. Direct traffic and referrals also contribute significantly, indicating that many visitors either bookmark our site or navigate from discussion forums and social media platforms. The dominance of organic search indicates that users are actively seeking open statistics about review processes and decisions. Notably, we also see a growing number of users referred from AI language models, including ChatGPT, Perplexity AI, Google Gemini, and DeepSeek.

**User Demographics**   Since its launch, Paper Copilot has naturally (no ads and marketing) attracted over 6 million impressions and one million site views globally, generating 4 million user-triggered events (e.g., clicks, scrolls) across 177 countries. The geographic distribution of users is visualized in Figure 1. These numbers reflect approximately 200,000 active users, with a maximum daily peak of 15,000 unique visitors. Over the past 28 days alone, the platform recorded 50,000 organic clicks from Google Search, highlighting strong and sustained community interest. These metrics underscore the importance of transparency in fostering engagement and demonstrate the community's enthusiasm for open and accessible systems.

## 4. Analysis

The collected traffic metrics and demographics reveal a global community that is not only aware of but also deeply invested in tracking review outcomes and statistics. In this section, we delve into the collected data to evaluate how different review models align with the community's demand for transparency and how they shape community's behaviors. We first clarify the primary modes of review disclosure, then assess the *community engagement* and validate *community interest*.

### 4.1. Review Disclosure

Many AI / ML venues have migrated from traditional closed platforms (e.g., Microsoft CMT) to more transparent platforms (e.g., OpenReview). However, as illustrated in Figure 2 and Figure 3, not all venues that move to OpenReview adopt a fully open process. We categorize venues into three disclosure modes:

- **Fully Open:** All reviews, discussions, and are publicly visible in real-time (e.g., ICLR).

- **Partially Open:** Reviews and discussions become public only after the decision phase concludes (e.g., NeurIPS, CoRL).

- **Fully Closed:** Reviews and discussions remain private indefinitely (e.g., ICML, CVPR).

Figure 3 shows that the *actual level of transparency* has remained mostly unchanged over the past decade, despite migrations to more flexible review platforms. Thus, while platform shifts suggest a trend toward openness, the community has not fully embraced complete real-time visibility.

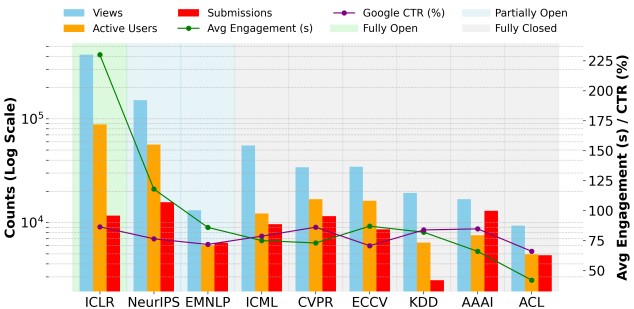

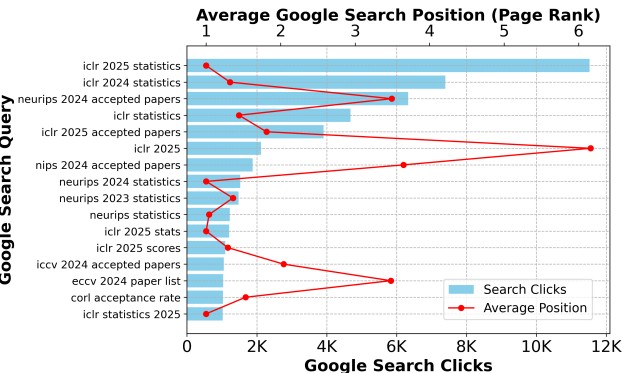

Figure 5: Comparison of views, active users (K = thousands), average engagement time (seconds), and Google Click-Through Rate (CTR, %) across venues, categorized by actual review disclosure (fully open, partially open, fully closed). The bar values are displayed on a **logarithmic scale** for better visibility of differences.

Figure 6: **Google Search Performance**: Search Clicks (K = thousands) and Average Page Rank Position for popular queries related to conference statistics and accepted papers.

## 4.2. Community Engagement

Before diving into the effective community interest, we first elaborate and understand who forms the community and how they engaged with open statistics. By analyzing key demographic markers—such as age, gender, and geographic distribution—we can better account for variations in usage patterns and guard against potential biases.

**Ages and Genders**  Figure 4a details user demographics by age and gender, revealing that the 18–24 age group accounts for the largest number of active users. Notably, younger males not only represent a substantial user base but also have the longest average engagement time (4 minutes 15 seconds), whereas older age brackets show a smaller user base and slightly shorter engagement durations (around 2.5 minutes). For females, engagement time remains relatively consistent across age groups, with a slight increase observed in the 65+ category (3 minutes 8 seconds). These findings suggest that early-career researchers—likely graduate students—are highly active and eager to follow review processes closely, making them potential drivers of future norms favoring transparency and standardization.

**Top 10 Countries**  Figure 4b displays both the number of active users and their average engagement time across ten countries. The United States and China lead with the largest user bases (60,648 and 59,269 users, respectively). However, locations with fewer total users, such as Singapore and Australia, exhibit notably high engagement times, exceeding 3 minutes on average. By contrast, the United Kingdom and Germany show comparatively shorter engagement (under 2 minutes), indicating distinct usage patterns. Taken together, these data not only confirm a global appetite for tracking AI / ML conference trends but also highlight the necessity for formal, widespread adoption of open-review principles that

can address the diverse needs of researchers worldwide.

## 4.3. Community Interests Validation

We quantize and validate community's activity and interests via various metrics including site visits, Google Organize Search Rankings and user activity on Openreview platform.

**Page Views and CTR**  Google Click-Through Rage (CTR) is the rate when an arbitrary user saw the site page via searching and made a click to it. As shown in Figure 5, the CTR remains consistently high across venues, with values ranging from 66.08% to 86.49%. This consistency suggests that researchers are equally curious about review statistics, irrespective of the conference's transparency level.

Based on this, Figure 5 demonstrates a significant disparity in engagement across conferences, largely influenced by their review modes. Notably, except for EMNLP, ACL, and KDD, submission numbers for most venues fall within a similar range of 11,000 to 15,000, providing a comparable baseline for analysis. Conferences adopting Fully Open or Partially Open review processes, such as ICLR and NeurIPS, exhibit substantially higher levels of community interaction compared to their Fully Closed counterparts. For example, ICLR, with its Fully Open review model, leads with 414,096 views, 88,220 active users, and an average engagement time of 3 minutes and 50 second—attracting nearly four times more views and six times more active users than NeurIPS (Partially Open) and far surpassing Fully Closed venues. In contrast, Fully Closed venues such as CVPR and ECCV lag significantly behind, with views under 35,000 and average engagement times of less than 1.5 minutes. This deeper metrics (page views and session duration) show that transparent conferences foster more sustained user involvement.

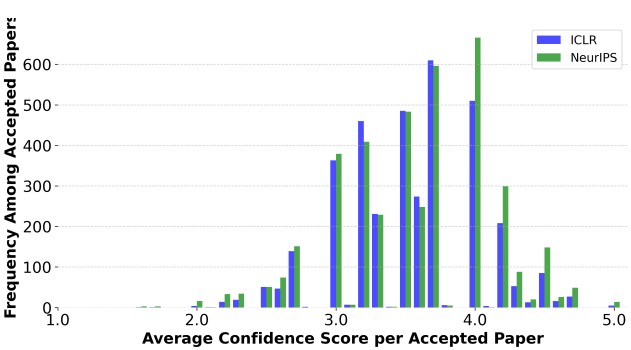

Figure 7: Distribution of Average Confidence Scores for Accepted Papers at ICLR and NeurIPS.

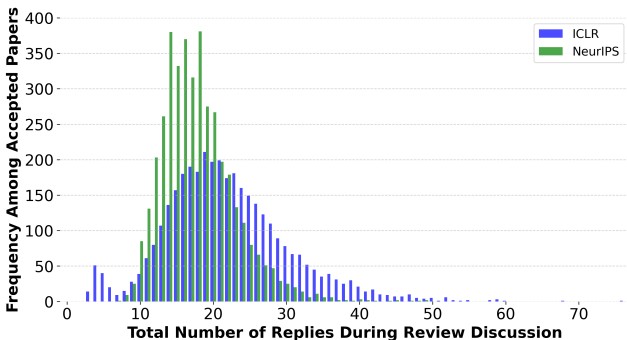

Figure 8: Distribution of the Total Number of Replies During Review Discussions for Accepted Papers at ICLR and NeurIPS.

**Organic Search Engine Rankings**   Figure 6 demonstrates a clear relationship between Google search clicks and the average position of pages for AI / ML-related queries, such as "ICLR 2025 statistics" and "NeurIPS 2024 accepted papers." Similar patterns are observed for Bing search metrics. These pages rank highly in search engine results, driven by algorithms like Google's PageRank (Page, 1999), which evaluates the quantity and quality of links a page receives from authoritative sources. High natural rankings for community-driven queries indicate that these pages effectively address the informational needs of users.

The prominence of pages related to open reviews and conference statistics underscores the AI / ML community's strong demand for transparency and accessibility in the research review process. The natural alignment between top-ranked content and community queries reflects a collective preference for systems that prioritize openness and accountability. This trend highlights the importance of the open review model as a mechanism to democratize access to research insights and foster trust in the peer review process, positioning it as a key expectation for the future of scientific publishing in AI / ML.

Additionally, the organic visibility of these resources highlights that many researchers—especially early-career individuals—actively seek centralized and transparent platforms. The consistent alignment between top-ranked content and user engagement demonstrates a grassroots push within the community for more accessible and open reviewing data, rewarding platforms that prioritize transparency with sustained attention and trust.

**OpenReview Dynamics**   Over recent years, confidence levels across review processes—fully open (e.g., ICLR), partially open (e.g., NeurIPS), and closed—have remained consistent, averaging between 3.5 and 3.6, as shown in Table 3. However, a closer look at the discussion data in 2024 comparing ICLR and NeurIPS in Figure 7 reveals a noteworthy distinction: ICLR exhibits a slightly lower concen-

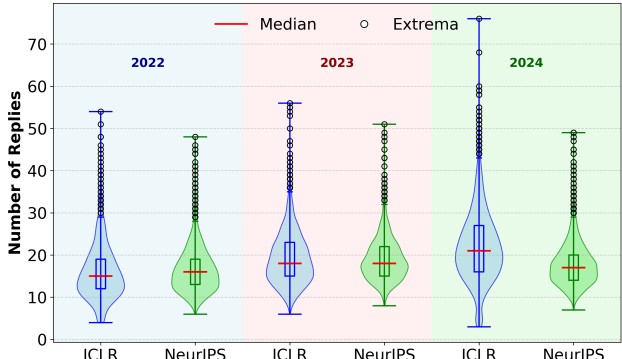

Figure 9: Violin plot of reply distributions for ICLR and NeurIPS accepted papers (2022–2024). The year reflects when discussions occurred, not the conference date.

| Year | ICLR | NeurIPS | ICML | CVPR |
|---|---|---|---|---|
| Modes | Fully Open | Partially Open | Fully Closed | Fully Closed |
| Source | API | API | Community | Community |
| 2022 | 3.52 ± 0.47 | 3.52 ± 0.53 | – | – |
| 2023 | 3.53 ± 0.49 | 3.52 ± 0.52 | – | – |
| 2024 | 3.53 ± 0.48 | 3.58 ± 0.54 | 3.54 ± 0.57 | 3.64 ± 0.48 |

Table 3: Review Confidence Statistics (Mean ± Std) for ICLR, NeurIPS, and CVPR (2022–2024). The year reflects when discussions occurred, not the conference date.

tration of high-confidence ratings among accepted papers. This may reflect the nature of open reviewing (Bharti et al., 2022), where public visibility fosters cautious, deliberate evaluations and mitigates overconfidence, contributing to a more thoughtful review process.

Discussion activity further underscores the advantages of fully open reviews. As shown in Figure 8, ICLR demonstrates a broader distribution of replies than NeurIPS in the same year, with a maximum count of 76 compared to 49 for NeurIPS, and significantly greater variance. The violin plot in Figure 9 confirms increasing medians and a

wider range of replies for ICLR across years, reflecting a more dynamic, iterative review environment. This vibrant engagement highlights the collaborative potential of open reviews, where authors, reviewers, and the community can engage in extended dialogues to refine research. By contrast, fully closed models restrict authors to a one-time rebuttal phase, limiting opportunities for clarification and broader community input.

These findings reinforce the value of fully open reviewing processes in fostering transparency, community engagement, and rigorous scrutiny. By enabling real-time, public discussions, open reviews systematically address ambiguities, encourage constructive feedback, and enhance reproducibility. As the demand for transparency and accountability in academic publishing grows, fully open processes offer a promising pathway to align peer review with these evolving standards.

## 5. Discussion: Close or Open

In this section, we examine three key challenges affecting the integrity of the fully closed peer review process and then propose how moving toward more open or partially open models could address these issues effectively.

### 5.1. Problems in Close Review

**Challenges for Younger Reviewers**  Demographic data indicate that a substantial portion of the AI research community now consists of younger individuals aged 18–24. As the field grows exponentially and the number of submissions soars, venues often face a shortage of qualified reviewers. In response, some venues (CVPR, 2025) require each submitting author to serve as a reviewer in order to manage the massive influx of papers.

While this policy helps alleviate reviewer shortages, it also compels younger, less-experienced researchers to evaluate work at the forefront of the field. Younger researchers are undoubtedly talented and growing in number, their limited familiarity with rigorous peer-review standards, combined with the pressure of large submission volumes, can lead to uneven or suboptimal feedback. This dynamic risks diluting the overall quality of the peer-review process.

A growing concern within the community form Paper Copilot highlights this issue: many authors report that reviewers struggle to fully understand the nuances of their submissions. While such claims are currently anecdotal and not yet quantifiable, future studies could analyze this trend systematically. As these reports continue to rise, they signal a potential systemic challenge that, if left unaddressed, could impose significant additional burdens on program committees, requiring extensive resources to mediate disputes and resolve misunderstandings stemming from insufficiently ex-

perienced reviewers.

**Ethical Concerns and AI Usage in Closed Review**
Whether closed or open, reviewers typically perform their duties with minimal oversight and must balance these tasks alongside their own research. The rise of large language models (LLMs) adds further complexity (Kuznetsov et al., 2024; Seghier, 2024; Zhang et al., 2022). Although LLMs can assist in revising or evaluating manuscripts, their unregulated use in a closed review context raises concerns about consistency and accountability.

In response, some venues have introduced policies to regulate LLM usage. However, enforcement remains challenging in a closed review environment, where the reviewing process—and any potential misuse—occurs largely out of public view. Moreover, these issues disproportionately affect younger reviewers, who may lack both the resources and the confidence to navigate potential ethical dilemmas. Overreliance on LLMs risks homogenizing feedback, thus reducing the diversity of perspectives that is vital for thorough peer review.

**Noticed Inconsistencies in Acceptance Records**  A notable concern emerging from closed-review venues is the discrepancy in author information between official conference records and final published versions. For instance, in 2024, some authors changed their names after paper acceptance, creating mismatches that are difficult to detect in a closed setting. While we refrain from revealing specific names or details to protect the authors' identities, these inconsistencies can be traced through publicly available statistics. Such incidents underline gaps in accountability and underscore the need for more robust regulatory mechanisms.

By contrast, open review processes naturally invite broader oversight, making it easier to spot and address potential irregularities. Publicly visible reviews and commentary foster collective accountability and discourage misconduct. Taken together, these observations highlight the urgent need for a more transparent and well-regulated review framework in the AI / ML community to maintain trust, ensure high-quality feedback, and safeguard research integrity.

### 5.2. Towards Open

The challenges described in prior sections underscore the urgent need for a more transparent and accountable review framework—one that supports the influx of younger reviewers, regulates AI usage, and preserves the integrity of scholarly discourse. Although expanding participation can bring fresh perspectives, it also risks undermining quality if newer reviewers lack structured mentorship and formal training. At the same time, ethical concerns regarding AI-assisted reviewing—such as homogenized feedback—illustrate the

fragility of closed systems, where limited oversight makes it difficult to enforce standards, detect biases, or reconcile inconsistencies in authorship records.

Moving toward open or partially open review processes offers a pragmatic solution to these issues. By making reviews publicly visible, community members can collectively scrutinize and address potential problems, from name-change discrepancies to excessive reliance on large language models. Such transparency fosters fairer evaluations, encourages ethical conduct, and cultivates a more collaborative environment for all participants. As AI research continues to evolve at a rapid pace, embracing open review mechanisms can help maintain a high standard of scholarly rigor while supporting the long-term credibility and vitality of the research community.

**User Studies**  To assess the research community's stance on open or partially open review processes, we conducted an interest survey prominently featured on Paper Copolit's front page. So far, the survey received over 228 responses, reflecting swift and enthusiastic engagement. Respondents spanned more than 20 distinct subfields—ranging from traditional AI / ML and robotics to medical informatics—covering a total of more than 50 major research venues.

When asked whether review scores should be publicly accessible at fully closed-review conferences such as CVPR 2025, 57% of respondents indicated they would be willing to share their scores with the community anonymously. This willingness points to growing support for more transparent peer-review practices. Equally notable was the speed with which respondents engaged, suggesting that the research community is eager to explore open or partially open review models that can address the challenges documented in this paper.

## 6. Alternative Views

While the preceding sections advocate for more transparent review processes, it is important to recognize that open or partially open systems are not without drawbacks. Critics highlight issues such as the potential for plagiarism, misappropriation of innovative ideas, and threats to proprietary research, raising valid questions about how best to balance openness with the need for confidentiality.

**Plagiarism**  One frequently cited concern is that open review may inadvertently facilitate plagiarism (Piniewski et al., 2024; Oviedo-García, 2024) if innovative concepts are publicly visible before a paper is formally published. When submissions are posted online (e.g., in open-review platforms or preprint servers like arXiv) and later rejected, these ideas remain accessible, allowing others to potentially adopt or iterate on them without proper attribution.

However, such issues are not exclusive to open review. In fact, the growing trend of researchers posting preprints on arXiv—regardless of whether a conference uses open or closed peer review—reveals that this challenge is part of a broader question of how to protect intellectual property in public forums.

Moreover, confidentiality can serve as a safeguard against idea theft, as it keeps manuscripts and reviews private until final decisions are made. This is seen as particularly important for early-career researchers and smaller institutions, which may lack the resources to compete if their concepts are exposed prematurely. **Yet, given the rapid pace of AI research and the prevalence of preprint culture, solutions to plagiarism concerns must extend beyond the open-versus-closed review debate.** The research community at large may need clearer norms, stronger protective measures, and more effective reporting systems to uphold ethical standards for all parties involved.

**Disclosure Policy**  For research scientists working at companies with patent-driven business models, such as those in the pharmaceutical, semiconductor, or AI industries, maintaining confidentiality in the peer review process is crucial. Many companies operate under strict intellectual property (IP) and patent disclosure policies to safeguard innovations before public release. Open review systems, which often require preprints or public sharing of submissions, could inadvertently expose proprietary research and jeopardize a company's ability to secure patents.

For example, in jurisdictions like the United States (USPTO, 2013), the first-to-file patent system requires that an invention must not have been publicly disclosed prior to filing. A submission shared in an open review process might qualify as prior art, rendering the invention unpatentable.

In these settings, critics of open review argue that confidentiality helps ensure that breakthroughs remain protected until the necessary legal steps are in place. Without this protection, competitors could quickly adopt or modify ideas, diluting the original innovator's advantage. While acknowledging the value of transparency, many researchers in industry and academia alike must balance the public benefit of sharing ideas with the practical need to safeguard proprietary innovations.

## 7. Conclusion

In this work, we analyzed the dynamics of open, partially open, and closed review processes in the AI/ML community, leveraging insights from Paper Copilot to highlight the growing interest in transparency. Our findings reveal that while fully open reviews promote transparency and engagement, they may also discourage reviewer confidence,

whereas closed systems lack accountability and broader community involvement. However, our analysis is limited by the rapid evolution of the AI/ML community, where shifting norms may outpace existing review models, and by potential biases in voluntary data submissions, which may not fully capture the community's diversity. Future work will focus on tracking the evolving dynamics and further expanding data, refining demographic analyses, and exploring peer review mechanisms further.

## Impact Statement

We offer a timely reflection on peer review practices in the AI / ML community and present actionable insights derived from large-scale community analytics. By openly sharing peer review metrics and fostering transparency-focused dialogue, we aim to empower early-career researchers, encourage broader community participation, and help shape the conversation around more accountable and inclusive review systems. We hope this work contributes to the development of future peer review policies that prioritize openness, fairness, and global accessibility—ultimately supporting a more equitable and trustworthy scientific ecosystem in AI / ML research.

## Acknowledgement

I would like to express my sincere gratitude to the global AI / ML research community for their trust, contributions, and engagement with Paper Copilot. Built entirely from scratch without institutional backing, this project was independently conceived, developed, and maintained during my spare time. I'm especially thankful to the many researchers who voluntarily shared data and insights, believing in the value of transparency and collective progress. I also thank my Ph.D. advisor, Prof. Yajie Zhao, for fostering an environment that supports academic freedom and self-directed exploration. While this work was carried out without direct involvement from my advisor or lab, her support allowed me the flexibility to pursue community-driven initiatives. All development, infrastructure, and operational costs were personally funded. More can be found at: https://papercopilot.com/acknowledgment/

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
