# OpenReview forum: "Position: The Artificial Intelligence and Machine Learning Community Should Adopt a More Transparent and Regulated Peer Review Process"
_ICML.cc/2025/Position_Paper_Track — ICML 2025 Position Paper Track poster_

### Official Review · Reviewer_BXnh · 2025-03-07

**Significance:** 2
**Argument Clarity:** 2
**Rating:** 2
**Confidence:** 3

**Questions:**

1) The choice of conference venues is unclear to me. Based on what criteria were these conferences selected?
2) The paper is entirely based on data collected via paper pilot. How/where was this collection initialized/advertised? To what extent does the user base reflect an unbiased sample of AI researchers and authors/reviewers of these conferences?
3) The position argues for a "more transparent and regulated peer-review process". Could you please elaborate on what "well-regulated" means (and how it can be differentiated from "transparent" and "open")?

**Discussion Potential:**

3

**Paper Summary:**

The paper analyses data from Paper Copilot to understand whether fully open-review processes (in contrast to partially open or closed) are better for community engagement.

**Position:**

Yes

**Position In Title:**

Yes

**Related Work:**

3

**Strengths And Weaknesses:**

**Strengths**
 + The topic is highly relevant and appropriate for a position paper
 + the paper is based on a large set of collected and publicly available data

**Weaknesses**
  * The discussion of engagement time, session duration, and page views and the conclusion that this relates to the openness of the review process are not convincing. There are many unknowns, e.g., is there the same amount of content on both websites?
  * Similarly, the discussion of open-review statistics of two conferences (ICLR and NeurIPS) is interesting, but there are many other differences, e.g., reviewer recruitment and assignment and length of the rebuttal period, topics, etc., so the conclusion (line 312/313) is again not convincing.
  * The alternative view section is rather vague, generic, and disconnected from the rest of the paper. It starts discussing IP and plagiarism issues that have not been mentioned anywhere before.

**Support:**

2

---

> ### Author Rebuttal · Authors · 2025-03-27
>
> We thank the reviewer for highlighting the relevance of the topic and the contribution of our data collection effort via Paper Copilot.
>
> > Q1. The conclusion that engagement time or page views relate to review process openness is not convincing; there are many unknowns.
>
> We agree that these metrics are not definitive indicators of review model effectiveness. In our paper, we present them as **proxy signals of community interest and engagement**, not as proof of causality.
>
> It’s important to note that **there is currently no large-scale public dataset** that allows for a rigorous, empirical comparison of different review models (e.g., open vs. closed). Most major conferences still operate with closed or partially open review models, and internal review data is not released in a form that supports cross-model evaluation.
>
> Therefore, we believe the only viable way forward is to **advocate for openness while simultaneously collecting data**. Over time, as venues like ICML and NeurIPS begin partially releasing public reviews post-decision, we will be able to **compare self-reported and official records** to validate patterns, quantify bias, and assess the trade-offs between models more rigorously.
>
> > Q2. ICLR vs NeurIPS comparison includes many confounders; therefore, the conclusions in line 312/313 are not convincing.
>
> Thank you for this point. We selected ICLR and NeurIPS for comparison based on their comparable size, topical scope, and availability of public data via OpenReview. However, we acknowledge differences in reviewer assignment policies, rebuttal formats, and timing. We will revise the text to qualify our conclusions and present them as exploratory rather than causal.
>
> > Q3. The alternative view section is vague and disconnected. IP and plagiarism are not mentioned earlier.
>
> Thank you for this feedback. We will revise the section to better connect concerns around plagiarism and IP with the broader discussion. In particular, we will clarify how concerns around plagiarism and IP are often cited as barriers to openness and how structured transparency—combined with protective mechanisms—can address these concerns.
>
> > Q4. The paper is entirely based on data collected via Paper Copilot. How was this initialized/advertised, and how representative is the user base?
>
> Paper Copilot’s growth has been mostly community-driven, with no advertising. The platform spread organically through search engines and researchers sharing it via Reddit, Twitter/X, and word-of-mouth. For the ICML 2025 review collection, over **528 entries were submitted within the first 24 hours, with 391 (∼74\%) including OpenReview or submission IDs**, enabling future validation against official public records.
>
> This organic interest has also been validated externally:
>
> * The platform **ranks highly in Google Search** for queries like “ICLR statistics.” or “ICML statistics.” Google's ranking algorithms prioritize content with **high engagement and low bounce rates**, providing indirect yet credible evidence of community value.
> * In this [ICML 2025 review discussion](https://www.reddit.com/r/MachineLearning/comments/1jiuqib/d_icml_2025_review_discussion/), an Area Chair noted that our collected review statistics consistently matched their internal batch.
> * Other active discussions are listed in the rebuttal to reviewer Bbc7.
> * We also collected institution-level data showing confirmed usage from researchers affiliated with top universities and institutes ([Affilations](https://papercopilot.com/))
>
> > Q5. What does “regulated” mean, and how is it different from “open” or “transparent”?
>
> * Open: making reviews, and author responses publicly visible.
> * Transparent: allowing the community to see and evaluate the peer review process (e.g., rebuttals, meta-reviews).
> * Regulated: implementing structured standards that help guide reviewer behavior and maintain consistency (e.g., checklists,  ethics oversight.
>
> A concrete example of regulation is ICML 2025’s introduction of a structured Claims and Evidence section in its Reviewer panels. This checklist prompts reviewers to assess the validity of claims, proofs, datasets, and experiments in a reproducible and consistent way.
>
>  > Q6. The choice of conference venues is unclear to me. Based on what criteria were these conferences selected?
>
> We selected venues based on two key criteria and will expand further to other venues.
>
> * High research visibility and influence, as indicated by the H5-index rankings on Google Scholar.
> * We prioritized conferences with **5,000 or more submissions per year**, where the impact of peer review practices is most significant—both in terms of review load and the number of early-career researchers affected.
>
> **If any part of our rebuttal remains unclear, we’re happy to clarify further. We appreciate your thoughtful review, and if our response resolves your concerns, we kindly ask you to consider updating your score. Thank you again for your feedback.**

---

> > ### Comment · Reviewer_BXnh · 2025-04-08
> >
> > Thank you for your effort and time. I appreciate that you will rephrase some of the conclusions as exploratory (rather than causal) since I believe ICML and NeurIPS are too different to draw conclusions solely based on the review process form. In addition, I acknowledge your effort to "advocate for openness while simultaneously collecting data" so I see this paper as a first step towards collecting data to understand the impact of review processes.
> >
> > I updated my score.

---

> > > ### Author Response · Authors · 2025-04-08
> > >
> > > Dear Reviewer BXnh,
> > >
> > > Thank you very much for taking the time to read our rebuttal and for updating your score—we truly appreciate your engagement and your thoughtful acknowledgment of our work as a first step toward understanding peer review processes through data.
> > >
> > > To clarify the scope of our data collection: over the past year, Paper Copilot has covered not only ICML and NeurIPS, but also a wide range of major venues across AI subfields—including ACL, KDD, ACM-MM, EMNLP, ECCV, CVPR, 3DV, and AAAI. These statistics pages represent community-driven efforts to aggregate and reflect peer review patterns across tracks and disciplines:
> > >
> > > * [ICML 2025 Statistics](https://papercopilot.com/statistics/icml-statistics/icml-2025-statistics/)
> > > * [ICML 2024 Statistics](https://papercopilot.com/statistics/icml-statistics/icml-2024-statistics/)
> > > * [CVPR 2025 Statistics](https://papercopilot.com/statistics/cvpr-statistics/cvpr-2025-statistics/)
> > > * [AAAI 2025 Statistics](https://papercopilot.com/statistics/aaai-statistics/aaai-2025-statistics/)
> > > * [NeurIPS 2024 Statistics](https://papercopilot.com/statistics/neurips-statistics/neurips-2024-statistics/)
> > > * [ACL 2025 Statistics](https://papercopilot.com/statistics/acl-statistics/acl-2025-statistics/)
> > > * [ACL 2024 Statistics](https://papercopilot.com/statistics/acl-statistics/acl-2024-statistics/)
> > > * [EMNLP 2024 Statistics](https://papercopilot.com/statistics/emnlp-statistics/emnlp-2024-statistics/)
> > >
> > > This year, we also initiated an effort to investigate the community’s willingness to further open their submitted scores. We believe this is a crucial next step—not only for advocating openness—but also for simultaneously collecting valuable data and learning how the community engages with transparency in practice.
> > >
> > > The response has been encouraging. These datasets have sparked wide discussion on social media and community forums, including Reddit and X, where related threads have reached millions of views:
> > > * [ICML Scores Interpretation](https://www.reddit.com/r/MachineLearning/comments/1bv8vbe/d_icml_scores_interpretation/)
> > > * [ICLR 2025 Paper Reviews Discussion](https://www.reddit.com/r/MachineLearning/comments/1gov5zd/d_iclr_2025_paper_reviews_discussion/)
> > > * [ICLR 2024 Decisions Are Coming Out Today](https://www.reddit.com/r/MachineLearning/comments/196uyub/d_iclr_2024_decisions_are_coming_out_today/?sort=new)
> > > * [ACL ARR June / EMNLP Review Discussion](https://www.reddit.com/r/MachineLearning/comments/1ebmas6/d_acl_arr_june_emnlp_review_discussion/)
> > > * [NeurIPS 2024 Paper Reviews](https://www.reddit.com/r/MachineLearning/comments/1efscr2/d_neurips_2024_paper_reviews/)
> > > * [X/Twitter Discussion](https://x.com/peter_richtarik/status/1842211968322560022) (375.8K views)
> > >
> > > We believe this is the first empirical effort to validate the research community’s interest in transparency and review process analysis at scale. During the ongoing ICML 2025 discussion phase alone, the ICML statistics page has already drawn over 46,754 visitors—further evidence of the appetite for better understanding and engagement around community interest.
> > >
> > > We also believe this aligns with the “Discussion Potential” criteria outlined in the [call for position papers](https://icml.cc/Conferences/2025/CallForPositionPapers): “The topic is likely to inspire constructive, useful discussion within the ICML community. The reviewer need not agree with the stated position.” While opinions may differ on the conclusions, we hope these empirical observations can serve as a catalyst for informed dialogue—something the ICML community is uniquely positioned to explore further.
> > >
> > > **Your evaluation and support play an important role in supporting this initiative**, and we hope to have the opportunity to present this work at ICML 2025 conference as a basis for broader community discussion. **If you find that our clarifications address your remaining concerns, we kindly ask you to consider updating your score to a weak accept**, giving this project the chance and opportunity to contribute meaningfully to the conversation at ICML 2025.
> > >
> > > Thank you again for your thoughtful review and consideration.

---

### Official Review · Reviewer_uuMu · 2025-03-13

**Significance:** 3
**Argument Clarity:** 3
**Rating:** 3
**Confidence:** 4

**Questions:**

1. Based on your analysis, do you think reviews should all be made more open? Or are you uncertain due to AI improvements? If so, what is the reason for your uncertainty? How would "shifting norms outpace existing review models"?
2. If you are uncertain, would it be better to maintain a diversity of review models?
3. How does review openness in AI/ML compare to other domains?

**Discussion Potential:**

3

**Paper Summary:**

This paper advocates for a more transparent, open, and standardized peer review process within AI and ML research, motivated by the rapid growth of conference submissions and the mixed success of fully open, partially open, and closed models. Drawing on usage data from Paper Copilot—which aggregates and analyzes conference statistics for a global audience of primarily early-career researchers—the authors highlight a strong appetite for openness, citing higher engagement and richer discussions at venues using open review. They emphasize, however, that younger reviewers and unregulated AI-driven evaluations pose quality and ethical challenges, and that concerns about plagiarism and corporate IP must still be addressed. Overall, their findings seem to suggest that the benefits of visibility, accountability, and collaboration outweigh the downsides, and they urge the community to adopt transparent processes with regulatory safeguards. However, they note that the "analysis is limited by the rapid evolution of the AI/ML community, where shifting norms may outpace existing review models".

**Position:**

Yes

**Position In Title:**

Yes

**Related Work:**

3

**Strengths And Weaknesses:**

Strengths:
- I applaud the authors building Paper Copilot and collecting empirical data to support their position. This is rare and highly appriciated!
- The paper addresses both practical (reviewer shortages, ethical AI usage) and structural (openness vs. IP constraints) issues in peer review.
- Discusses potential solutions and acknowledges ongoing debates around confidentiality, plagiarism, and corporate IP—showing awareness of real-world constraints.

Weaknesses:
- The authors are reluctant to take a strong position and couch their findings in the concluding sentences. Even if the rapid evolution of AI impacts the author's position, it seems that this position is much less useful if made in this way. Ultimately, conferences must select a review process and it would be helpful to have a clear position supported by empirical data.
- If AI progress leads to uncertainty, perhaps a plurality of review models is superior as a diversification strategy for our community. The authors statements seems to support this (advocating for an open process but saying this may not be future-proof) but they don't discuss it explicitly.
- It may be useful to compare AI/ML review procedures to other disciplines. My sense is that AI/ML is more open, however, but an empirical perspective could be helpful.

**Support:**

4

---

> ### Author Rebuttal · Authors · 2025-03-26
>
> We thank the reviewer for their encouraging feedback and for recognizing the effort to ground our position in empirical data via Paper Copilot.
>
> > Q1. The authors are reluctant to take a strong position and couch their findings in the concluding sentences. Ultimately, conferences must select a review process and it would be helpful to have a clear position supported by empirical data.
>
> We appreciate this observation. While our initial language was cautious due to the field's rapid evolution, we agree that the paper benefits from a stronger stance. In the revision, we will make it clear that we advocate for a more open and transparent peer review process by default. Exceptions (e.g., IP-sensitive or safety-critical submissions) should be narrowly scoped and justified.
>
> We also highlight ICML 2025 as an example of positive development toward regulation, where the newly added Claims and Evidence checklist in the reviewer instructions guides reviewers through more structured and consistent evaluations. We believe such checklists serve as effective regulatory tools and should be adopted more broadly across venues.
>
> > Q2. Based on your analysis, do you think reviews should all be made more open? Or are you uncertain due to AI improvements? If so, what is the reason for your uncertainty?
>
> We do advocate for a more open review process overall, but we acknowledge that this transition should be **phased and mindful of legitimate exceptions** (e.g., protecting whistleblowers, corporate IP, or sensitive topics). Our uncertainty stems not from reluctance about openness, but from the lack of **publicly available comparative data** on how different review models perform in practice.
>
> That is exactly why we launched Paper Copilot before the shift toward openness began in earnest—so that we could track longitudinal data and measure community sentiment and patterns **before**, **during**, and **after** these changes occur.
>
> > Q3. If you are uncertain, would it be better to maintain a diversity of review models?
>
> We fully agree that **diversity of review models is beneficial** for the community, especially during this transitional period. However, a major challenge is that the current AI/ML ecosystem lacks sufficient public data to evaluate which models are most effective. That’s one of the key reasons we launched the Paper Copilot project **prior to widespread adoption of openness**, so that we could begin building a foundation for future comparisons.
>
> From a venue’s perspective, it is difficult to compare review models internally, since all authors participate in the same review pipeline regardless of the model (open vs. closed). As submission volume rapidly increases, we believe the **openness or closeness of review models deserves greater attention**.
>
> Paper Copilot serves as a **complementary perspective** that collects community-level signals and user-submitted review data across conferences and models. Over time, this will allow for broader comparisons, validation, and bias correction—something that is currently not feasible within most individual venues.
>
> > Q4. How would “shifting norms outpace existing review models”?
>
> We appreciate this question. By “shifting norms,” we refer to the rapid changes in reviewer demographics (more early-career researchers), the growing use of AI tools (e.g., LLMs for reviewing), and new publication formats (e.g., artifact evaluations, code requirements). These shifts may outpace current review infrastructures, which were originally designed under different assumptions about expertise, volume, and visibility.
>
> We argue that openness and regulation are important mechanisms to keep the review system adaptable and accountable as norms evolve.
>
> > Q5. How does review openness in AI/ML compare to other domains?
>
> We appreciate this question. While the primary focus of this paper is on the AI/ML domain, we have also observed interest in peer review practices across other disciplines—evidenced by over 200+ Google Forms ([Let me know your interest survey](https://docs.google.com/forms/d/e/1FAIpQLScLmmhRITQCwyNLLb_DzXBH4n3CsSOtKvPhwIRRmq8mOIJwVg/viewform?usp=sf_link)) submissions expressing curiosity beyond AI/ML. In the revised paper, we will include brief comparisons to select fields such as HCI, medicine, optics, and radiology to provide additional context. Nonetheless, we believe AI/ML is uniquely positioned to lead a broader shift toward transparency, given its large, engaged online community and infrastructure like OpenReview that supports open and iterative review models.
>
> **If any aspect of our response remains unclear, we’d be happy to clarify. We appreciate your thoughtful review and hope our rebuttal addresses your concerns. If so, we kindly invite you to consider revising your score. Thank you again for your feedback.**

---

### Official Review · Reviewer_FiBW · 2025-03-14

**Significance:** 3
**Argument Clarity:** 3
**Rating:** 4
**Confidence:** 4

**Questions:**

On page 6, the authors state that *"ICLR demonstrates a broader distribution of replies than NeurIPS"* and argue that this *"reflects a more dynamic, iterative review environment."* The question here is: why is *a broader distribution of replies* considered inherently beneficial? Would a higher number of replies generally be regarded as a positive factor? The logical connection in this argument could be strengthened. More broadly, while the paper's main claim—that the peer review process should be more open and transparent—is reasonable and aligns with ongoing efforts to improve research integrity and reviewer-author communication, the reasoning could be more rigorously substantiated. The reviewer agrees that openness can enhance transparency and help reviewers better understand authors' intentions, which is beneficial for the research community. However, the paper would be more solid if it provided more concrete evidence or a more complete logical chain to substantiate the claim.

**Discussion Potential:**

3

**Paper Summary:**

This paper examines peer review models in the AI/ML community, comparing open, partially open, and closed systems. The study highlights that while fully open reviews enhance transparency and foster engagement, they may also reduce reviewer confidence, whereas closed systems provide privacy at the cost of accountability and broader community participation.

## update after rebuttal

No further concerns.

**Position:**

Yes

**Position In Title:**

Yes

**Related Work:**

3

**Strengths And Weaknesses:**

#### Strengths:
- As a researcher in the ML field, the reviewer really appreciate the authors' contribution to the community through the launch of PaperCopilot.
- The paper is highly informative, offering a comprehensive discussion on different peer review models. It systematically analyzes the advantages and drawbacks of fully open, partially open, and closed review systems, supported by engagement data from PaperCopilot.

#### Minor Issues:
- The title on each page should be "ICML 2025" rather than "ICML 2024."

**Support:**

2

---

> ### Author Rebuttal · Authors · 2025-03-26
>
> We thank the reviewer for their thoughtful comments and for recognizing both the informativeness of our analysis and the value of the Paper Copilot platform.
>
> > Q1. Why is a broader distribution of replies considered inherently beneficial? Would a higher number of replies generally be regarded as a positive factor?
>
> We appreciate this critical question. In the paper, we used the distribution of reviewer replies (e.g., more multi-round exchanges at ICLR compared to NeurIPS) as a **proxy for the level of dialogue between reviewers and authors**. We acknowledge that a higher number of replies does not automatically imply higher review quality; however, we believe such iterative exchanges can reflect a **more participatory and responsive review process**, especially when combined with transparency. We will revise the text to emphasize this nuance and frame the broader distribution as a useful signal—not a definitive metric.
>
> > Q2. The paper's reasoning is reasonable but would be stronger if supported by more concrete evidence or a more complete logical chain.
>
> We agree, and we will revise the submission to more clearly articulate the connection between transparency, reviewer accountability, and community trust. While our current evidence is based on observational data, as more venues release public reviews, we will be able to compare our community-driven metrics with official records to validate and strengthen our findings. For example, during ICML 2025, we received over 528 entries within the first 24 hours after the release of the review data to authors, with 391 (~74%) including an OpenReview or submission ID. These identifiers will allow us to validate self-reported scores against the official public reviews once they are disclosed publicly.
>
> > Q3. ICML 2025 instead of ICML 2024
>
> Thank you—we will fix this in the updated version.

---

### Official Review · Reviewer_Bbc7 · 2025-03-15

**Significance:** 3
**Argument Clarity:** 4
**Rating:** 3
**Confidence:** 4

**Questions:**

See Strengths And Weaknesses

**Discussion Potential:**

4

**Paper Summary:**

The authors argue the need of a more transparent, open, and regulated peer review process in the AI/ML community. It highlights the challenges posed by the exponential growth in submissions to top-tier conferences and the limitations of current peer review models (fully open, partially open, and fully closed). The authors analyze data from "Paper Copilot," a platform aggregating AI/ML conference statistics, to demonstrate the community's interest in transparency and the benefits of open review systems. The paper provides evidence of increased engagement and collaboration in open review models while addressing challenges such as reviewer biases, ethical concerns, and the need for regulatory guidelines. It also discusses alternative views, including concerns about plagiarism and intellectual property risks, and proposes solutions to balance transparency with confidentiality.

**Position:**

Yes

**Position In Title:**

Yes

**Related Work:**

3

**Strengths And Weaknesses:**

pro:
The paper examines the strengths and weaknesses of different review models (fully open, partially open, and fully closed) and provides a balanced analysis of their implications for transparency, accountability, and community involvement.

con:
Although the paper calls for a regulated peer review process, it provides limited details on what such regulations would entail or how they could be enforced effectively.

The paper data analysis is based on "Paper Copilot" data which is valuable but the data is uploaded voluntary (regardless of ICLR) which is a biased distribution.

**Support:**

3

---

> ### Author Rebuttal · Authors · 2025-03-26
>
> We thank the reviewer for their thoughtful summary and recognition of our effort to analyze and compare different peer review models.
>
> > Q1. Although the paper calls for a regulated peer review process, it provides limited details on what such regulations would entail or how they could be enforced effectively.
>
> Thank you for this helpful comment. We agree that more clarity is needed on what “regulated” means in this context. In the revised version, we will clarify that “regulation” refers to **structured standards that help guide reviewer behavior and maintain consistency**. ICML 2025 provides a strong example of this shift by introducing a more detailed Claims and Evidence section in their [Reviewer Instructions](https://icml.cc/Conferences/2025/ReviewerInstructions), which includes specific criteria and a checklist for evaluation. This section provides a checklist for evaluation—e.g., “Are the claims supported by clear and convincing evidence?”—which moves away from traditional free-form structures seen in venues like ICLR 2025, NeurIPS 2024, and CVPR 2024.
>
> As the field continues to scale and more early-career reviewers are recruited, these structured guidelines become essential for maintaining review quality. We believe such mechanisms represent a practical and scalable form of “regulation” and advocate for their adoption across conferences.
>
> Additionally, one observation we would like to raise is that the author list during the review process and the final version on the conference website may differ. While this issue may not fall strictly under review regulations, it does represent a potential loophole in the transparency pipeline that deserves further community discussion.
>
> > Q2. The paper data analysis is based on "Paper Copilot" data which is valuable but the data is uploaded voluntarily (regardless of ICLR) which is a biased distribution.
>
> We appreciate this important observation. Voluntary data collection does have limitations, and we do not claim it to be unbiased. This topic has been actively discussed in the community. For example, we conducted real-time data collection for ICML 2025 ([ICML 2025 Statistics - Paper Copilot](https://papercopilot.com/statistics/icml-statistics/icml-2025-statistics/)), and community discussions also reflect interest and engagement with the platform. In the same thread, an Area Chair commented that their internal batch matched the collected statistics consistently ([discussion](https://www.reddit.com/r/MachineLearning/comments/1jiuqib/comment/mjn07dy/?utm_source=share&utm_medium=web3x&utm_name=web3xcss&utm_term=1&utm_content=share_button)).
>
> Additional community discussions that reflect Paper Copilot’s ongoing relevance include:
> * [ICML Scores Interpretation](https://www.reddit.com/r/MachineLearning/comments/1bv8vbe/d_icml_scores_interpretation/)
> * [ICLR 2025 Paper Reviews Discussion](https://www.reddit.com/r/MachineLearning/comments/1gov5zd/d_iclr_2025_paper_reviews_discussion/)
> * [ICLR 2024 Decisions Are Coming Out Today](https://www.reddit.com/r/MachineLearning/comments/196uyub/d_iclr_2024_decisions_are_coming_out_today/?sort=new)
> * [ACL ARR June / EMNLP Review Discussion](https://www.reddit.com/r/MachineLearning/comments/1ebmas6/d_acl_arr_june_emnlp_review_discussion/)
> * [NeurIPS 2024 Paper Reviews](https://www.reddit.com/r/MachineLearning/comments/1efscr2/d_neurips_2024_paper_reviews/)
>
> We view it as a necessary approach in the absence of publicly released review data. Fully closed or partially open review venues rarely release full review data, making voluntary data collection a mechanism required to estimate community-level insights. As venues like ICML and NeurIPS begin releasing (opt-in) reviews post-decision, **we can begin comparing collected reviews against publicly released reviews, which allows us to quantify and reduce bias over time**. For instance, in ICML 2025, we received over 528 entries within the first 24 hours, and **391 (∼74\%) of them provided their OpenReview or submission IDs**. These IDs will allow us to directly compare reported data against the official public reviews on release.
>
> To reduce bias, we must (1) continuously track self-reported data, and (2) wait for more venues to adopt open or partially open models. With venues like ICML 2025 beginning to release reviews for accepted papers and opt-in rejected ones ([source](https://icml.cc/Conferences/2025/PeerReviewFAQ)), we can begin comparing collected scores with official releases. This will help validate and correct biases over time. Therefore, I believe that the only way forward is to advocate for more transparency and continue collecting data in parallel. While this effort may take years, it is central to the long-term mission of the Paper Copilot project.
>
> **We’re happy to provide further clarification if needed. Thank you for your careful review—if our response has resolved your concerns, we would sincerely appreciate your consideration in updating the score.**

---

### Decision · Program_Chairs · 2025-04-30

**Decision:**

Accept (poster)

**Comment:**

This paper advocates for a more transparent and regulated peer review process. This is supported by evidence from Paper Copilot, a popular website which aggregates data from review processes in ML.

Reviewers generally felt that this offered a comprehensive look at different types of open review models and how they affected the process. Some of the proposed weaknesses were thoughtful questions, which I consider as supportive of the case for this paper's interest to the community. Perhaps the biggest weakness may be that the data is inherently biased, which the authors acknowledge, but this is truly the best we have at this point, so whatever insights we get might still be valuable.

Note: the running title is "Submission and Formatting Instructions for ICML 2024" which should be changed to this paper's title.  It also suggests that the authors used the 2024 style files.  Ensure you have the 2025 style files and update the paper accordingly.